# MSC.sTRAIL Has Better Efficacy than MSC.FL-TRAIL and in Combination with AKTi Blocks Pro-Metastatic Cytokine Production in Prostate Cancer Cells

**DOI:** 10.3390/cancers11040568

**Published:** 2019-04-21

**Authors:** Andrea Mohr, Tianyuan Chu, Greg N. Brooke, Ralf M. Zwacka

**Affiliations:** 1Cancer and Stem Cell Biology Group, School of Biological Sciences, University of Essex, Colchester CO4 3SQ, UK; tc17896@essex.ac.uk; 2Molecular Oncology Group, School of Biological Sciences, University of Essex, Colchester CO4 3SQ, UK; gbrooke@essex.ac.uk

**Keywords:** mesenchymal stem cells, cell therapy, sTRAIL, prostate cancer, AKT, AKTi, IL-6, CXCL5, ENA-78

## Abstract

Cell therapy is a promising new treatment option for cancer. In particular, mesenchymal stem cells (MSCs) have shown potential in delivering therapeutic genes in various tumour models and are now on the verge of being tested in the clinic. A number of therapeutic genes have been examined in this context, including the death ligand TRAIL. For cell therapy, it can be used in its natural form as a full-length and membrane-bound protein (FL-TRAIL) or as an engineered version commonly referred to as soluble TRAIL (sTRAIL). As to which is more therapeutically efficacious, contradicting results have been reported. We discovered that MSCs producing sTRAIL have significantly higher apoptosis-inducing activity than cells expressing FL-TRAIL and found that FL-TRAIL, in contrast to sTRAIL, is not secreted. We also demonstrated that TRAIL does induce the expression of pro-metastatic cytokines in prostate cancer cells, but that this effect could be overcome through combination with an AKT inhibitor. Thus, a combination consisting of small-molecule drugs specifically targeting tumour cells in combination with MSC.sTRAIL, not only provides a way of sensitising cancer cells to TRAIL, but also reduces the issue of side-effect-causing cytokine production. This therapeutic strategy therefore represents a novel targeted treatment option for advanced prostate cancer and other difficult to treat tumours.

## 1. Introduction

The tumour necrosis factor related apoptosis-inducing ligand (TRAIL), also known as Apo2L, CD253 or TNFSF10, can induce apoptosis in cancer cells while sparing normal cells. The molecular basis for the tumour-selective activity of TRAIL remains to be fully defined [1,2,3,4,5,6]. Unlike conventional chemotherapeutics, TRAIL induces apoptosis in a p53-independent manner and without substantial non-specific cellular or DNA damage [7]. Since p53 is frequently inactivated in human tumours, TRAIL is therefore able to induce apoptosis in cancer cells that are normally hard to treat. These features make TRAIL a promising agent for anti-cancer therapy. 

TRAIL is a member of the TNF superfamily, which forms multimers that interact with cognate receptors on the cell surface [8,9,10]. It binds to four membrane-bound death receptors and one soluble receptor, TRAIL-R1/DR4, TRAIL-R2/DR5, TRAIL-R3/ DcR1, TRAIL-R4/DcR2 and osteoprotegerin (OPG). TRAIL-R1 and TRAIL-R2 possess a conserved cytoplasmic region termed the death-domain (DD) that is needed for TRAIL-induced apoptosis [11]. Following binding of TRAIL, a protein complex, known as death-inducing signalling complex (DISC) consisting of TRAIL-R1 and/or TRAIL-R2, the adaptor protein Fas-associated death domain (FADD) and procaspase-8, is formed.

In the DISC, procaspase-8 is activated by a mechanism that involves dimerisation and proteolytic cleavage [12,13,14]. In contrast, TRAIL-R3, TRAIL-R4 and the soluble receptor OPG have inhibitory functions, because they lack the intracellular death-domains, and are therefore regarded as decoy receptors [8,10,15,16,17,18]. Various mechanisms have been described for the action of these decoy receptors. While TRAIL-R3 prevents the assembly of the DISC, TRAIL-R4 disrupts the formation of ligand-independent pre-assembled homotrimeric complexes of agonistic TRAIL-receptors [19]. Further, heterotrimers containing TRAIL-R3 are less active in transmitting apoptosis signals following binding of TRAIL [20,21,22]. 

Activated caspase-8 sets in motion the apoptosis cascade via a number of mechanisms; firstly, it can directly cleave executioner caspases such as procaspase-3 to the active form of caspase-3. This leads to apoptosis by proteolytic cleavage of a series of cellular targets giving rise to hallmarks of apoptosis such as cellular shrinkage, chromatin condensation and nuclear fragmentation [23]. Alternatively, or in addition, caspase-8 can cleave Bid (BH3-interacting domain death agonist), a proapoptotic protein from the Bcl-2 family, to a truncated form known as tBid [24]. This tBid then interacts with the Bcl-2 family proteins, Bax and Bak, leading to conformational changes and insertion of these factors into the outer mitochondrial membrane [25,26,27]. There, the pro-apoptotic proteins are believed to form pores giving rise to mitochondrial outer membrane permeabilisation (MOMP). MOMP results in the release of cytochrome c, Smac/DIABLO and other pro-apoptotic factors into the cytosol. Cytochrome c amplifies the apoptotic signal emitted by activated caspase-8 via additional caspase-9 activation in the apoptosome, while Smac/DIABLO releases the X-linked inhibitor of apoptosis protein (XIAP) block on caspase-9 and executioner caspases, thereby also strengthening apoptosis [28,29,30].

Even though clinical studies with recombinant TRAIL (rTRAIL) and agonistic antibodies to TRAIL-receptors have shown that they are safe, only moderate evidence of therapeutic efficacy has been observed [31,32,33]. Despite frequent high-dose injections, insufficient bioavailability at the site of tumour growth has been identified as one of the reasons for lack of efficacy [31,32,33,34,35]. In order to overcome this problem, gene and cell therapy approaches utilising TRAIL have been tested and shown to be effective in experimental tumour models [36,37,38,39,40,41]. In particular, approaches using mesenchymal stem cells (MSCs) have proven to be successful in this context [42,43,44,45]. There are principally two different options to deliver TRAIL, either as a full-length and membrane bound protein, reflecting the normal genetic sequence of the *TNFSF10* gene, or as an engineered version including the ectodomain of TRAIL (aa114–281) and a strong signal peptide that ensures effective secretion [42,46]. 

As MSC-based delivery of TRAIL is about to be tested in clinical trials, it is important to identify optimal versions of TRAIL that have the best potential for therapeutic efficacy. Therefore, we compared cells expressing full-length TRAIL (FL-TRAIL) or soluble TRAIL (sTRAIL) in different experimental systems and approaches to investigate their capacity to induce cancer cell killing. Furthermore, we analysed how different forms of TRAIL affect the production of potentially side-effect-causing cytokines [47,48,49], and how this problem could be overcome by testing different sensitisation approaches in TRAIL resistant prostate cancer cells.

## 2. Results

### 2.1. Comparison of sTRAIL and FL-TRAIL 

TRAIL is a 281 amino-acid long type-II membrane protein. However, when used experimentally as a recombinant protein, only the soluble ectodomain (usually aa114–281) is expressed and purified. In cell therapeutic applications, it is possible to use either the full-length, membrane-bound version (FL-TRAIL) or engineer cells to secrete a smaller, soluble form (sTRAIL). Our goal was to compare the cell death inducing activities of the two TRAIL types in the context of cell therapy, and to investigate how other non-apoptotic TRAIL-signalling pathways and outcomes were affected. The FL-TRAIL expression construct consisted of the TRAIL cDNA (aa1–281) under the control of the CMV promoter (Figure 1a). For the sTRAIL construct, the TRAIL ectodomain was fused to an Isoleucine Zipper (ILZ) for trimerisation, the signal peptide of the human *Fibrillin* gene to provide effective secretion, and a Furin cleavage site to release the ILZ-sTRAIL protein into the extracellular space (Figure 1a). 

Both constructs were transfected into HEK293 cells and a western blot with respective whole cell protein extracts showed FL-TRAIL and sTRAIL running at the expected different molecular weights (Figure 1b). These results indicate that in sTR−AIL expressing cells a substantial amount of sTRAIL still resides inside of the cells or is associated with the membrane. To further explore this, we carried out flow cytometric analyses of the cells and found similar TRAIL signals on the surface of FL-TRAIL and sTRAIL expressing cells suggesting that significant levels of sTRAIL are displayed on the cell membrane before it is cleaved (Figure 1c). Thus, using sTRAIL in cell therapeutic applications potentially combines the advantages of FL-TRAIL, achieving high TRAIL concentrations focused around the vicinity of the cell therapy vehicle with the wider reaches of the secreted molecule. Finally, we measured the TRAIL concentrations in supernatants of FL-TRAIL and sTRAIL expressing 293 cells. After filtration through a 0.45 μm filter, the supernatants were applied to a TRAIL ELISA. The results demonstrate that only sTRAIL expressing cells gave rise to detectable TRAIL levels in the cellular supernatants (Figure 1d). Our findings that FL-TRAIL is not secreted or released are in contrast to some previous reports [50,51]. Therefore, we sought to find out the possible reasons behind this discrepancy.

### 2.2. Apoptotic Cells Release TRAIL-Containing Cellular Fragments Mimicking Secretion

First, we analysed how different methods of clearing the cell supernatant of cell debris affected the results of the TRAIL ELISA. We transfected HEK293 cells with the FL-TRAIL and sTRAIL constructs, respectively. Then, either the unprocessed crude supernatants, cleared supernatants by sedimentation of cell debris through centrifugation, or cleared supernatants by filtration through a 0.45 μm filter were applied to a TRAIL ELISA. FL-TRAIL gives rise to cellular particles containing TRAIL that can be detected in the supernatant. Centrifugation, to some degree, and filtration completely removes these particles and the TRAIL signal, whereas sTRAIL is not affected by these clearing methods (Figure 2a).

It is hypothesised that basal, transfection- or TRAIL-induced apoptosis led to the generation of TRAIL-containing cellular fragments from dead cells that were detected in the FL-TRAIL samples. To test this, TRAIL resistant CHO cells (Figure 2b) were used for the production of the two different TRAIL versions (Figure 2c). After transfection and processing of the supernatants in the described manner the samples were again applied in their crude, centrifuged or filtered form to a TRAIL ELISA. Importantly, even crude supernatants from FL-TRAIL transfected CHO cells contained almost no detectable TRAIL (Figure 2d), indicating that indeed TRAIL-induced apoptosis in HEK293 cells (and possibly other TRAIL-sensitive producer cells) can result in the appearance of TRAIL in the supernatant. It seems that the detection is caused by TRAIL-containing debris from apoptotic cells present in the culture. The TRAIL signal can be removed by centrifugation and filtration demonstrating it is not proteolytically cleaved and secreted, in contrast to the engineered sTRAIL. Next, we wanted to confirm these findings in highly relevant MSCs as they hold great promise as a cell therapeutic delivery vehicle in cancer.

### 2.3. MSC.sTRAIL Exhibits More Potent Effects on 2D- and 3D-Cultures of Cancer Cells than MSC.FL-TRAIL

We used adenoviral vectors expressing FL-TRAIL or sTRAIL to transduce mouse and human MSCs, generating MSC.FL-TRAIL and MSC.sTRAIL. Subsequently, we analysed crude, centrifuged or filtered culture supernatants by TRAIL ELISA. There was no detectable level of TRAIL in any of the supernatants from the FL-TRAIL samples, whereas sTRAIL was secreted to substantial levels (Figure 3a,b). This could also be achieved in different types of MSCs, including those derived from adipose tissue, bone marrow, and umbilical cord (Figure 3c). The fact we did not observe any TRAIL in the supernatant of the FL-TRAIL samples can be explained by their resistance to TRAIL (Figure 3d). Thus, in principle MSCs are good cell therapy vehicles for TRAIL, as they do not succumb to their own molecular armoury. 

We went on to further test and compare MSC-produced FL-TRAIL with sTRAIL by testing them on the human colorectal cancer cell line HT-29. The results show that the supernatants of MSC.sTRAIL caused a decrease in survival (Figure 3e) and a corresponding increase in apoptosis (Figure 3f), whereas the MSC.FL-TRAIL supernatants exerted no effects on HT-29 cells. These results were also confirmed by transwell co-cultures (Figure 3g,h). Mixing of HT-29 cells with MSC.FL-TRAIL or MSC.sTRAIL, at a ratio of 10:1, resulted in an increase in cell death in both cases, but with a significantly greater effect for MSC.sTRAIL (Figure 3i,j). Next, MSC.FL-TRAIL and MSC.sTRAIL were tested on a second colorectal cancer cell line, Colo205. Again, while MSC.FL-TRAIL induced apoptosis in this line, the effect was far greater with MSC.sTRAIL (Figure 3k). In conclusion, MSC-delivered sTRAIL appears to be superior compared to FL-TRAIL when mixed with cancer cells, and MSC.FL-TRAIL showed no evidence of TRAIL secretion. Therefore, we went on to test MSC.sTRAIL on more difficult to treat cancer cells.

### 2.4. Docetaxel can Sensitise TRAIL-Resistant Prostate Cancer Cells to TRAIL-Induced Apoptosis

Prostate cancer cells are resistant to TRAIL-induced apoptosis (Figure 4a), but several ways have been described to sensitise cells to this molecule [52]. From a therapeutic standpoint it is beneficial to choose combinations that include clinically used treatments or specifically targeted drugs. For prostate cancer the regular chemotherapy is docetaxel or related compounds [53]. Docetaxel is part of the chemotherapy drug class of taxanes and is a semi-synthetic analogue of Taxol, which is extracted from the bark of the rare Pacific yew tree. Docetaxel is produced from a chemical, derived from the European yew tree which is more plentiful and renewable [54]. We used three different prostate cancer cell lines to test docetaxel and TRAIL combinations: PC3, LNCaP and C4-2B. PC3 cells were established from bone metastasis of a grade IV prostate cancer patient. LNCaP cells are androgen-sensitive human prostate adenocarcinoma cells derived from lymph node metastasis. C4-2B cells are a more aggressive derivative subline of LNCaP cells and were generated by passaging through nude mice. 

When we co-treated these prostate cancer cells lines with docetaxel and rTRAIL we found that all three could be sensitised to TRAIL-induced apoptosis (Figure 4b–d). The effect was most prominent in PC3 and LNCaP cells (Figure 4b,c), whereas C4-2B cells (Figure 4d) exhibited higher docetaxel sensitivity and hence less sensitisation effect. Thus, TRAIL/docetaxel co-treatment might be an option to increase the efficacy of both agents and reverse TRAIL resistance in prostate cancer and other cancer types. Docetaxel did not lead to an increase in TRAIL-receptor surface expression (Appendix A) as many other chemotherapeutic drugs do [55,56]. Notwithstanding, the treatment regime can make these tumour types amenable to MSC.sTRAIL treatment. However, it has recently been reported that TRAIL and/or chemotherapeutic drugs can also have detrimental effects in the form of cytokines being induced in response to treatment [48,57]. Therefore, we tested whether TRAIL induces cytokines in TRAIL-resistant prostate cancer cells and whether docetaxel can repress this potential induction.

### 2.5. TRAIL Induces CXCL5/ENA-78 and IL-6 in Prostate Cancer Cells, which cannot be Blocked by Docetaxel Co-treatment

When we treated PC3 cells with rTRAIL and applied the medium supernatant to a cytokine antibody array, containing antibodies against 42 cytokines, we found CXCL5/ENA-78 and IL-6 to be up-regulated when compared to untreated cells (Figure 5a and Appendix A). CXCL5/ENA-78 and IL-6 are cytokines that have been associated with tumour progression and have therefore the potential to cause serious side effects [58]. TRAIL-induced CXCL5/ENA-78 expression was confirmed by ELISA, which demonstrated that it is increased in a TRAIL-dose dependent manner (Figure 5b). Similarly, an IL-6 ELISA demonstrated that addition of TRAIL to PC3 cells increased IL-6 levels in the supernatant in a dose dependent manner (Figure 5c). MSC-produced sTRAIL also led to CXCL5/ENA-78 (Figure 5d) and IL-6 (Figure 5e) induction as measured by ELISA. When we mixed CHO cells expressing FL-TRAIL or sTRAIL with PC3 cells, CXCL5/ENA-78 was not induced (Figure 5f), whereas the IL-6 levels were increased (Figure 5g). Although we used CHO cells in these experiments to avoid interference from MSC-derived factors, these results point to a potential benefit of cell-delivered TRAIL. However, to address a potential worst-case scenario we continued to use rTRAIL and sTRAIL to investigate whether co-treatment with docetaxel could prevent the TRAIL-dependent induction of the two cytokines. Co-treatment of rTRAIL and docetaxel, previously shown to enhance apoptosis (Figure 4), was not able to block the increased production of either CXCL5/ENA-78 (Figure 5h) nor IL-6 (Figure 5i). Similarly, sTRAIL from MSCs also induced CXCL5/ENA-78 (Figure 5j) and IL-6 (Figure 5k), but docetaxel again could not inhibit this increase in cytokine production (Figure 5j,k).

Given the potential problems these cytokines could cause, other combination treatments were investigated that promote significant TRAIL-induced apoptosis and at the same time block or limit the concomitant up-regulation of pro-metastatic factors such as CXCL5/ENA-78 and IL-6.

### 2.6. AKTi can Sensitise Prostate Cancer Cells to TRAIL and Block Cytokine Production

Many prostate cancer cells harbour mutations in the *PTEN* gene [59]. The protein encoded by *PTEN* is a phosphatase negatively regulating the levels of phosphatidylinositol-3,4,5-trisphosphate thereby dampening the pro-growth PI3K and AKT (also known as PKB) signalling pathways [60]. Thus, prostate cancer cells with *PTEN* mutations or deletions have elevated PI3K and AKT/PKB activation levels, which is likely to bestow survival benefits onto these cells when challenged by physiological surveillance mechanisms or treatment with chemotherapeutic agents or biologicals such as TRAIL. Therefore, rTRAIL/PI3K inhibitor (PI3Ki) and rTRAIL/AKT-inhibitor (AKTi) combinations were tested on PC3, LNCaP and C4-2B cells, all bearing mutations/deletions of PTEN [61]. The results show that both the AKTi and PI3Ki approaches led to significant sensitisation to TRAIL-induced apoptosis in all three prostate cancer cell lines, with the exception of the PI3Ki in PC3 cells (Figure 6a–c). 

Overall, better results were achieved with AKTi and therefore the AKTi was subsequently tested in combination with MSC.FL-TRAIL and MSC.sTRAIL. We also confirmed that AKTi was functional in PC3 cells at the basal level and also when cells were treated with TRAIL and docetaxel (Appendix A). In the cell-mixing experiments, including AKTi, we found MSC.sTRAIL gave rise to increased apoptosis in prostate cancer cells, while MSC.FL-TRAIL exerted only marginal effects (Figure 6d). 

Next, we examined the cell death-inducing activities of MSC.FL-TRAIL and MSC.sTRAIL in two prostate cancer cell 3D models that more closely resemble the three-dimensional nature of a tumour in vivo. Again, the superior effects of MSC.sTRAIL in combination with AKTi was confirmed compared to MSC.FL-TRAIL (Figure 6e,f). 

Finally, we wanted to know whether AKTi would be able to halt the TRAIL-induced up-regulation of CXCL5/ENA-78 and IL-6. Under conditions that result in apoptosis sensitisation and reduced survival (Figure 6g), AKTi almost completely blocked the TRAIL-induced increase in CXCL5/ENA-78 (Figure 6h), whereas the elevated levels of IL-6 were reduced by approximately 30% (Figure 6i). Hence, AKTi specifically inhibits the pathways leading to CXCL5/ENA-78 and reduces IL-6 production. Similarly to rTRAIL, sTRAIL from MSCs also induced CXCL5/ENA-78 (Figure 6j) and IL-6 (Figure 6k) levels. Co-treatment with AKTi inhibited the increase in CXCL5/ENA-78 production (Figure 6j) and significantly reduced the increase in IL-6 (Figure 6k). A triple treatment with rTRAIL, AKTi and Docetaxel did not result in any further improvements in cytokine repression (Appendix A). In conclusion, MSC.sTRAIL acts better in combination with AKTi than MSC.FL-TRAIL on prostate cancer cells with PTEN mutations/deletions. Hence, this combination can not only achieve a cancer-specific sensitisation to TRAIL-induced apoptosis, but also concurrently reduces the levels of potentially tumour-progressive cytokines such as CXCL5/ENA-78 and IL-6.

## 3. Discussion

Despite disappointing results from the initial clinical trials, the death ligand TRAIL remains a promising candidate for use in anti-cancer therapies, as it is able to trigger cancer cell death through its two apoptosis-inducing receptors, TRAIL-R1 and TRAIL-R2 [34,41]. Targeting cancer cells through receptors on their surface can be advantageous as it provides a good degree of accessibility and is the underlying principle of several targeted therapies such as Trastuzumab/Herceptin or Bevacizumab/Avastin [62,63,64,65]. Better delivery methods, for example with cells that possess tumour-infiltrating properties, is a potential approach to afford better therapeutic effects of TRAIL [66,67]. In this context, MSCs have shown great promise in preclinical studies, and clinical trials with MSCs delivering TRAIL are imminent [68].

For therapies that utilise recombinant TRAIL it is clear that only the extracellular domain of the protein is needed. For cell therapeutic approaches TRAIL can be displayed as a full-length membrane protein or produced as an engineered secreted form. Both versions have advantages and disadvantages. Using membrane-bound TRAIL limits the apoptosis induction to the immediate vicinity of the delivering cells. This restricts detrimental effects arising from high levels of systemic TRAIL, but also the therapeutic impact as highlighted by Luetzkendorff et al. [69]. They found that a substantial number of MSCs expressing FL-TRAIL was required to yield an anti-tumour response. This was achieved by direct-tumour-injections, whereas systemic application of their FL-TRAIL carrying MSCs had no effect on the growth of colorectal cancer xenografts owing to a low rate of tumour integration of MSCs. This is in contrast to several studies with MSCs producing sTRAIL in various cancer models [42,46,56,70,71]. However, another report noted that MSC-delivery of FL-TRAIL is superior to soluble TRAIL for cancer therapy [51]. 

In the present study, we observed that MSCs producing sTRAIL have more potent effects than cells expressing FL-TRAIL, both in conventional cell culture assays as well as 3D tumour cell spheroids. This is despite membrane-bound FL-TRAIL being a very strong apoptosis inducer during direct cell interactions, but for cell therapeutic applications the wider reaching activities of secreted and diffusing sTRAIL appear to be more important and effective [72]. We found no evidence for FL-TRAIL being processed and released from the membrane, thus its activity is indeed limited to the immediate vicinity of the cell therapy vehicle. Instead we discovered that cellular fragments containing TRAIL, originating from dead cells, can be mistaken for secreted TRAIL. The fact that we could eliminate the TRAIL signal in supernatants of FL-TRAIL expressing cells by filtration through a 0.45 μm filter rules out exosomes/extracellular vesicles (EVs) as the carrier of TRAIL, because they would not have been excluded by the filtration step, given their size of 40–100 nm [73]. Another type of extracellular vesicle, so called microvesicles (MVs), can be up to 1 μm in diameter, but also smaller vesicles (100 nm) may bud from the cell surface to form MVs [74]. Thus, it is unlikely that all TRAIL containing MVs would have been eliminated by filtration. Additionally, when we used cells, that were resistant to TRAIL-induced apoptosis, including MSCs, we could no longer detect TRAIL in the supernatants of FL-TRAIL expressing cells. Together these results suggest that cellular fragments from apoptotic cells are the source of the detected TRAIL. 

In relation to the discrepancies in the observed therapeutic effects of MSC.sTRAIL vs. MSC.FL-TRAIL, they are reconcilable by the fact that in lung cancer models MSCs with FL-TRAIL might work very well, as most systemically administered MSCs will infiltrate the lungs in the first 24–48 h, before they are cleared and appear in other tissues including tumours, but in far smaller numbers [75]. Thus, MSC.FL-TRAIL might be sufficient for lung cancer treatment, but not for other cancer types. Accordingly, the pending “TACTICAL” clinical trial for lung cancer with MSC-TRAIL vehicles will be using full-length TRAIL [76]. 

Irrespective of the form of TRAIL used, the fact that not all tumours are naturally sensitive to TRAIL remains an issue. This not only limits the therapeutic effects, but also allows cancer cells to respond with induced cytokine production. Some of these factors are known to have pro-metastatic activity and could cause serious side effects. It was shown that TRAIL-triggered cytokine secretion from TRAIL resistant cancer cells is FADD- and caspase-8 dependent. CXCL1, CXCL5, CCL2, and IL-8 were significantly induced by TRAIL in lung, colorectal and pancreatic cancer cell lines [48,77]. In particular CCL2 was responsible for monocyte polarisation to myeloid-derived suppressor cells (MDSCs) and M2-like macrophages. Blocking CCL2 signalling reduced the TRAIL-induced tumour-supportive immune microenvironment and tumour growth. MDSCs can influence tumour progression in a number of ways. They are directly implicated in the promotion of metastasis development by participating in the formation of a pre-metastatic niche, promoting angiogenesis and tumour cell invasion [78]. In addition, tumour associated macrophages (TAMs), and in particular M2-like macrophages have been associated with early tumour dissemination and progression [79,80]. These findings point to potential problems with TRAIL-based cancer therapies such as increased cancer cell proliferation and tumour progression [48]. Similarly, signalling by FasL/CD95L, another member of the death ligand family, was found to be associated with the production of an array of cytokines and chemokines, including IL-6, IL-8, CXCL1, MCP-1, and GM-CSF [81]. 

We found that rTRAIL and sTRAIL induced CXCL5/ENA-78 and IL-6 expression in TRAIL-resistant prostate cancer cells. CXCL5/ENA-78 is a member of the CXC chemokine family and is known to be activated by inflammatory cytokines such as interleukin-1 or tumour necrosis factor-alpha [82]. It is an ELR+ chemokine that can activate CXCR2 and has been described to promote angiogenesis [58]. In addition, the CXCR2/CXCL5 pair has been shown to enhance tumour progression by increasing the formation, recruitment and suppressive activity of MDSCs [83]. IL-6 is an interleukin that acts as a pro-inflammatory cytokine, but can also have anti-inflammatory activities as a myokine, i.e., when it is activated in response to muscle contraction during exercise for example [84,85]. IL-6 has multiple effects on tumour progression, of which some are the result of direct action on tumour cells, while others are the result of its activity on other cells in the tumour microenvironment [86]. It has a direct growth stimulatory effect on many tumour cells by activating several signalling pathways including Ras/Erk and STAT-3 [87,88]. IL-6 can also play an important role in promoting metastasis formation in distal organs by increasing the expression of bFGF, MMP-2, and VEGF, which contribute to invasion and angiogenesis [89]. Additionally, it has been reported that IL-6 in primary tumours promotes tumour self-seeding, i.e., the re-colonisation of the tumour of origin by circulating tumour cells. This process is able to accelerate tumour growth, angiogenesis, and stromal cell recruitment [90], and self-seeding might be an important contributor to tumour aggressiveness [91]. Taken together these results suggest that TRAIL can induce cytokines with possible detrimental effects, particularly when a number of cytokines are concomitantly upregulated and acting with complementarity. Interestingly, CHO cells expressing FL-TRAIL or sTRAIL, when mixed with PC3 cells, induced IL-6 expression whereas CXCL5/ENA-78 remained unchanged. While we do not have a mechanistic explanation for this effect, the results point to important benefits of TRAIL delivered by cell therapy that are worthy to be studied in more detail in the future.

The prostate cancer cells utilised in this study, as with many other tumour types, were resistant to TRAIL and therefore needed a sensitising treatment. This provides the opportunity to combine a TRAIL-based therapy with standard chemotherapy, which would be good practice for testing a new experimental cell therapy such as MSC.sTRAIL. Docetaxel is the most commonly used chemotherapy for men with advanced prostate cancer and synergies with TRAIL have been reported [92]. In this paper, the authors suggested that the TRAIL sensitisation by Docetaxel was mediated mainly by phosphorylation of BCL2 by activated JNK. Alternatively, a treatment targeted for the specific cancer type could be utilised. Advanced prostate tumours, including the PC3 cells that we examined, harbour mutations/deletions in *PTEN* that lead to high constitutive PI3K and AKT activation, providing a major therapeutic opportunity in the form of PI3Ki and AKTi [93]. Indeed, activation of the AKT survival pathway was demonstrated to contribute to TRAIL resistance and the PI3K inhibitor LY294002, or knockdown of AKT, sensitised resistant cancer cells to TRAIL [94]. While both, docetaxel and AKTi approaches worked to increase the TRAIL responsiveness, only a combination treatment of TRAIL with AKTi blocked the production of TRAIL-induced cytokines.

Thus, MSC-delivered sTRAIL in combination with AKTi (or other appropriate co-treatments) is a potential future therapeutic approach for advanced prostate cancer and other difficult to treat tumour types, as it enables TRAIL to induce apoptosis and prevents or reduces the production of harmful cytokines at the same time.

## 4. Materials and Methods 

### 4.1. Cell Culture and Reagents

All chemicals, unless otherwise stated, were purchased from Sigma (St. Louis, MO, USA). Recombinant TRAIL (rTRAIL) was from R&D Systems (Minneapolis, MN, USA). Docetaxel, AKTi (MK-2206) and PI3Ki (LY294002) were from Stratech (Stratech Scientific, Ely, UK).

The colorectal carcinoma cell lines Colo205 (obtained from Dr. Eva Szegezdi) and HT-29 (ATCC, Manassas, VA, USA) were cultured in RPMI (Lonza, Basel, Switzerland) and in McCoy’s 5A (Lonza), respectively. HEK293 cells were grown in DMEM (Lonza) and CHO cells (ATCC) in Ham’s F12. The prostate cancer cell lines, LNCaP (ATCC), C4-2B (obtained from Dr. Wafa Al-Jamal) and PC3 (ATCC), were cultured in RPMI medium (Lonza). All media were supplemented with 10% fetal bovine serum (FBS) (Thermo Fisher Scientific, Waltham, MA, USA), 100 U/mL penicillin and 100 μg/mL streptomycin.

Human bone marrow derived MSCs were from Lonza, human adipose derived stem cells were from Amsbio (Cambridge, MA, USA) and human umbilical cord derived MSCs were from Promocell (Heidelberg, Germany). The different human MSCs were cultured in StemMACS MSC Expansion Medium (Miltenyi Biotec, Bergisch Gladbach, Germany). Murine MSCs were isolated and cultured as previously described [75]. In experiments where it was stated that MSCs were used, but not further defined, human bone marrow derived MSCs were used.

### 4.2. Generation of TRAIL Constructs

The TRAIL full-length (FL-TRAIL) cDNA was cloned by RT-PCR from Jurkat cell RNA [39]. The sTRAIL segment (aa114–281) was amplified by PCR from the FL-TRAIL construct as described previously [95]. Recombinant E1/E3-deleted adenoviral vectors expressing sTRAIL and FL-TRAIL (Ad.sTRAIL, and Ad.FL-TRAIL), from the cytomegalie virus (CMV) promoter/enhancer elements were generated using the ViraPower adenoviral expression system (Thermo Fisher Scientific) as described earlier [44]. Transductions with adenoviral vectors were carried out as described before [96] at an MOI of 200 if not described otherwise. Adenovirus harbouring the LacZ gene coding for the bacterial β-galactosidase gene was used as a control.

### 4.3. Transwell co-Cultures

TRAIL expressing MSCs were co-cultured with HT-29 cells in 24-well plates, where HT-29 cells in the lower chamber are separated from the MSCs in the upper chamber by a semipermeable membrane (0.4 μm pore size). For this, MSCs were seeded in a 6-well plate and transduced with Ad.sTRAIL or Ad.FL-TRAIL respectively. 48 h later, the MSCs were harvested and transferred into the chamber of the transwell-insert (Sarstedt, Nümbrecht, Germany). For further analyses the transwell-inserts were removed.

### 4.4. 3D Spheroid Colorimetric Proliferation/Viability Assay 

3D Spheroid Colorimetric Proliferation/Viability Assay (Trevigen, Gaithersburg, MD, USA) was performed according to the manufacturer’s instructions. Briefly, upon completion of spheroid formation, the LNCaP spheroids were co-treated with 1 μM AKTi, 5 ng sTRAIL or an equal volume of FL-TRAIL secreting cell supernatant for 3 days before cell viability was assessed. 

### 4.5. 3D Culture

500 cells were resuspended in 40 μL Matrigel (Beckton Dickinson, Franklin Lakes, NJ, USA). The 3D culture was established for 5 days before the tumoroids were co-treated with 1 μM AKTi and TRAIL expressing MSCs. Control vector transduced MSCs served as control. Images were taken after 3 days, before cell viability was determined by resazurin reduction.

### 4.6. Crystal Violet Assay

Cells were seeded at a density of 1 × 10^4^ cells in a 24-well plate. After 24 h the cells were treated for 3 days. Crystal violet staining was performed as previously described [57]. The crystal violet stain was solubilised in methanol and the optical density of each well was measured at 570 nm. The average OD570 of non-stimulated cells was set to 100%. 

### 4.7. ELISA

CHO and 293 cells were Fugene-HD (Promega, Fitchburg, WI, USA) transfected with pcDNA.FL-TRAIL and pcDNA.sTRAIL. 48 h post-transfection, the supernatants were removed and filtered through a 0.45 μm syringe filter (Sarstedt) unless otherwise stated. Secreted TRAIL levels were determined by DuoSet ELISA (Biotechne, Minneapolis, MN, USA), according to the manufacturer’s instructions. 1 ng sTRAIL, or an equal volume of FL-TRAIL cell supernatant, was used in further assays unless otherwise stated.

For the IL-6 (Duoset, Biotechne) and CXCL5/ENA-78 ELISA (Duoset, Biotechne), PC3 cells were seeded and treated with 5 ng TRAIL or 1 ng sTRAIL unless otherwise stated. For combination treatments, the cells were pre-treated with 10 μM AKTi for 10 min and with 12.5 nM Docetaxel for 6 h before TRAIL was added for a further 72 h. 

CHO cells were transfected with an empty pcDNA3 vector (ctrl), pcDNA.FL-TRAIL and pcDNA.sTRAIL. After 24 h, cells were washed and trypsinised, then counted and mixed with PC3 cells at a ratio of 1:5 for 72 h. The resulting supernatants were analysed for CXCL5/ENA-78 and IL-6 by ELISA.

### 4.8. Cell Surface Staining and FACS Analyses 

Cells were analysed for TRAIL and TRAIL-receptor expression on their cell surface by flow cytometry. 1 × 10^6^ cells per test were washed with PBS supplemented with 2% bovine serum albumin. Anti-human TRAIL antibody (Diaclone, Besançon, France), anti-human TRAIL-R1 (Diaclone) and anti-human TRAIL-R2 (Diaclone) were added for 20 min at 4 °C. Following a washing step, the cells were incubated with rat anti-mouse IgG1 PE (BioLegend, San Diego, CA, USA). An isotype-matched immunoglobulin was used as control. After fixation with 2% paraformaldehyde samples were immediately analysed.

### 4.9. Apoptosis Assays

For apoptosis measurements, cells were treated 24 h with TRAIL, unless otherwise stated. In combination treatments with docetaxel, cells were pre-treated with docetaxel for 48 h before TRAIL was added. Apoptosis was determined 24 h after TRAIL was added. In combination treatments with AKTi, cells were co-treated with 1 μM AKTi and TRAIL for 24 h. TRAIL expressing cells were harvested 48 h post transfection/transduction and added to the tumour cells at the indicated ratios. Apoptosis was determined 24 h later.

DNA hypodiploidy staining: cells including their medium supernatant and PBS wash were harvested and centrifuged at 1300 rpm for 7 min at 4 °C. After washing with PBS, Nicoletti buffer (Sodium citrate 0.1% (*w*/*v*) containing 0.1% Triton X-100 (*w*/*v*) and propidium iodide 50 μg/mL) [97,98] was added to the cell pellets, tubes were vortexed for 10 s at medium speed and left for 1 h in the dark (4 °C). The fluorescence intensity was then measured in a flow cytometer and analysed. Cell death was also determined by flow cytometry by measuring changes in the forward (FSC) and side scatter (SSC).

### 4.10. Western Blot 

Proteins were separated by SDS-PAGE and transferred onto PVDF membranes (GE Healthcare Biosciences, Piscataway, NJ, USA). Primary and secondary antibodies were diluted in TBS, 0.1% Tween and 3% BSA. Bands were visualised with ECL Western blotting substrate (Thermo Fisher Scientific). Rabbit anti-TRAIL (Peprotech, Rocky Hill, NJ, USA), rabbit anti-phospho AKT and rabbit anti-AKT (Cell Signaling Technology, Danvers, MA, USA) were used as primary antibodies and peroxidase-conjugated anti-rabbit antibody (Santa Cruz, Dallas, TX, USA) was used as secondary antibody.

### 4.11. Cytokine Array

For these studies, supernatants from PC3 cells treated with 5 ng rTRAIL were used. They were compared to the cytokine levels of untreated PC3 cells. The respective supernatants were diluted 10-fold and applied to human cytokine antibody arrays III (RayBio, Norcross, GA, USA) according to manufacturer’s instructions. Each cytokine signal was normalised to the internal control and the fold upregulation in response to rTRAIL was calculated.

### 4.12. Statistical Analysis

Experimental values are expressed as mean value ± standard error (SE). For significance analyses, analysis of variance (ANOVA) between groups was used.

## 5. Conclusions

Our results demonstrate that in the context of cell therapy an engineered, secreted version of TRAIL has more anti-cancer potency than full-length TRAIL that remains membrane-bound and is not secreted into the extracellular space. Therefore, in most cases the use of sTRAIL will be more beneficial. Furthermore, TRAIL resistant prostate cancer cells can be sensitised by AKTi co-treatment, which also prevents the TRAIL-induced induction of tumour promoting and pro-metastatic cytokines from the cancer cells, providing an added advantage. This response is specific, as it cannot be afforded by docetaxel co-treatment, despite being able to sensitise prostate cancer cells to TRAIL-induced apoptosis. Thus, MSC-delivered sTRAIL in combination with AKTi is a potential option for advanced prostate cancer with *PTEN* mutations/deletions, but other combinations may be necessary in a cancer-specific, or even patient-specific, manner.

## Figures and Tables

**Figure 1 cancers-11-00568-f001:**
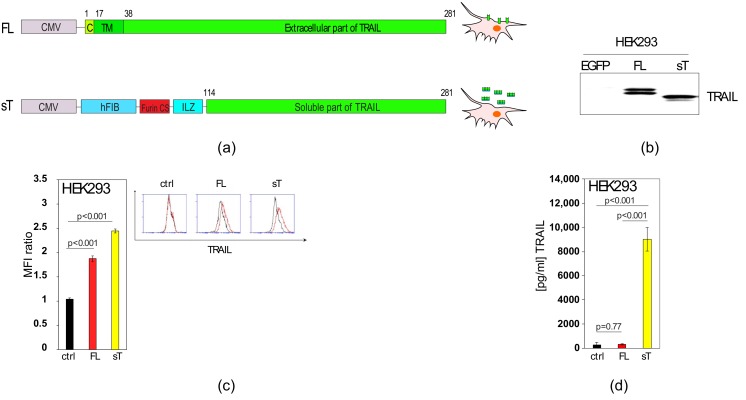
FL-TRAIL and sTRAIL are expressed in HEK293 cells, but only sTRAIL is secreted into the supernatant. (**a**) Schematic depiction of full length, membrane bound TRAIL (FL) and soluble TRAIL (sT) expression cassettes including depiction of the localisation of the two TRAIL forms when expressed in cells. The full-length version is the TRAIL cDNA corresponding to aa1-aa281 containing a cytoplasmic part (C), transmembrane region (TM) and the extracellular domain. The sTRAIL construct consists of a hFIB heterologous signal peptide, a Furin cleavage site (Furin CS), an Isoleucine Zipper (ILZ) and the sTRAIL part from aa114–281. Both constructs are under the control of the CMV promoter within pcDNA3 expression plasmids or adenoviral vectors. (**b**) HEK293 cells were transfected with pCDNA3 constructs for EGFP, FL-TRAIL (FL) or sTRAIL (sT). The resulting protein lysates were western blotted and probed with a TRAIL antibody. (**c**) HEK293 cells were transfected with an empty pCDNA3 plasmid (ctrl), as well as constructs for FL-TRAIL (FL) and secreted TRAIL (sT), respectively. The cells were then stained with a TRAIL antibody followed by a secondary antibody carrying a PE fluorescent tag and analysed by flow cytometry. (**d**) HEK293 cells were transfected with expression constructs for FL-TRAIL (FL), secreted TRAIL (sT) or an empty plasmid (ctrl). After 48 h the supernatants were filtered through a 0.45 μm filter and the resulting filtrates used for a TRAIL ELISA. Values represent mean ± SE.

**Figure 2 cancers-11-00568-f002:**
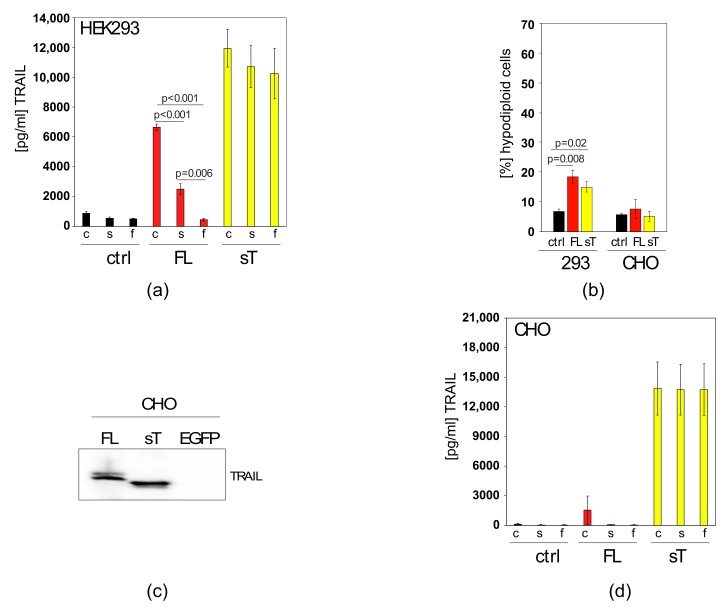
In apoptosis-sensitive, FL-TRAIL expressing cells, TRAIL appears in the supernatant owing to apoptosis, but not secretion (**a**) HEK293 cells were transfected with an empty plasmid (ctrl), a FL-TRAIL construct (FL) or an sTRAIL (sT) construct. The resulting medium supernatants were either taken as crude supernatants (c), centrifuged (s) or filtered through a 0.45 μm filter (f) before TRAIL levels were measured by ELISA (**b**) Apoptosis was measured in HEK293 and CHO cells that were transfected with an empty plasmid (ctrl), an FL-TRAIL construct (FL) or an sTRAIL construct (sT). (**c**) CHO cells were transfected with pCDNA3 constructs expressing FL-TRAIL (FL), sTRAIL (sT) or EGFP. The resulting protein lysates were subjected to western blotting. The membrane was then probed with a TRAIL antibody. (**d**) CHO cells were transfected with an empty plasmid (ctrl), a FL-TRAIL construct (FL) or an sTRAIL (sT) construct. The resulting medium supernatants were either taken as crude supernatants (c), centrifuged (s) or filtered through a 0.45 μm filter (f) before TRAIL levels were measured by ELISA. Values represent mean ± SE.

**Figure 3 cancers-11-00568-f003:**
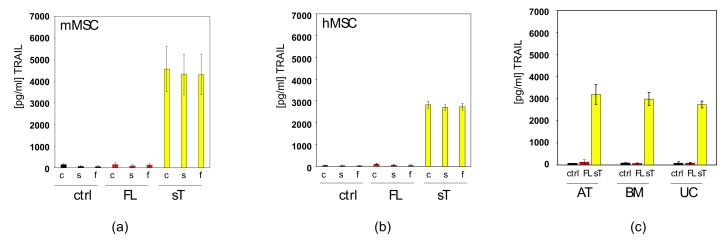
MSC.sTRAIL induces higher apoptosis levels in colorectal cancer cells than MSC.FL-TRAIL. (**a**) Murine MSCs (mMSCs) were transduced with a control adenoviral vector (ctrl), a vector expressing FL-TRAIL (FL) or a vector expressing sTRAIL (sT). The resulting medium supernatants were either taken as crude supernatants (c), centrifuged (s) or filtered through a 0.45 μm filter (f) before TRAIL levels were measured by ELISA. (**b**) Human MSCs (hMSCs) were transduced with a control adenoviral vector (ctrl), a vector expressing FL-TRAIL (FL) or a vector expressing sTRAIL (sT). The resulting medium supernatants were either taken as crude supernatants (c), centrifuged (s) or filtered through a 0.45 μm filter (f) before TRAIL levels were measured by ELISA. (**c**) Different types of MSCs (adipose tissue-derived - AT, bone marrow-derived - BM, and umbilical cord-derived - UC) were transduced with a control adenoviral vector (ctrl), a vector expressing FL-TRAIL (FL) or a vector expressing sTRAIL (sT). The resulting medium supernatants were passed through a 0.45 μm filter before TRAIL levels were measured by ELISA. (**d**) Apoptosis was measured in different types of MSCs, adipose tissue-derived (AT), bone marrow-derived (BM) and umbilical cord-derived (UC) and mouse (mMSC) MSCs that were transduced with a control adenoviral vector (ctrl), a vector expressing FL-TRAIL (FL) or a vector expressing sTRAIL (sT). (**e**) Survival was measured in HT-29 cells after exposure to supernatants from MSC.FL-TRAIL (red) and MSC.sTRAIL (yellow) at increasing concentrations of sTRAIL. For MSC.FL-TRAIL supernatants equal volumes of supernatant were used for each corresponding sTRAIL concentration. (**f**) Apoptosis was measured in HT-29 cells after exposure to supernatants from MSC.FL-TRAIL (red) and MSC.sTRAIL (yellow) at increasing concentrations of sTRAIL. For MSC.FL-TRAIL supernatants equal volumes of supernatant were used for each corresponding sTRAIL concentration. (**g**) Survival was measured in HT-29 cells that were co-cultured with control MSCs (ctrl), MSC.FL-TRAIL (FL) or MSC.sTRAIL (sT) in 24-well plates fitted with transwell tissue culture inserts. (**h**) Apoptosis was measured at 24 h and 72 h in HT-29 cells. For this HT-29 cells were co-cultured with control MSCs (ctrl), MSC.FL-TRAIL (FL) or MSC.sTRAIL (sT) in 24-well plates fitted with transwell tissue culture inserts. (**i**) Morphologically altered cells were measured by flow cytometry (changes in FSC/SSC) after mixing HT-29 cells with control MSCs (ctrl), MSC.FL-TRAIL (FL) or MSC.sTRAIL (sT) at a ratio of 10:1. (**j**) Apoptosis measurements of HT-29 cells mixed with control MSCs (ctrl), MSC.FL-TRAIL (FL) or MSC.sTRAIL (sT) at a ratio of 10:1. (**k**) Apoptosis measurements of Colo205 cells mixed with control MSCs (ctrl), MSC.FL-TRAIL (FL) or MSC.sTRAIL (sT) at different ratios (5:1; 10:1; 100:1). Representative images of Colo205 cells mixed with control MSCs (ctrl), MSC.FL-TRAIL (FL) or MSC.sTRAIL (sT) are shown on the right side of the panel. MSCs are pointed out by blue arrows, apoptotic cells by red arrows. Scale bar is 30 μm. Values represent mean ± SE.

**Figure 4 cancers-11-00568-f004:**
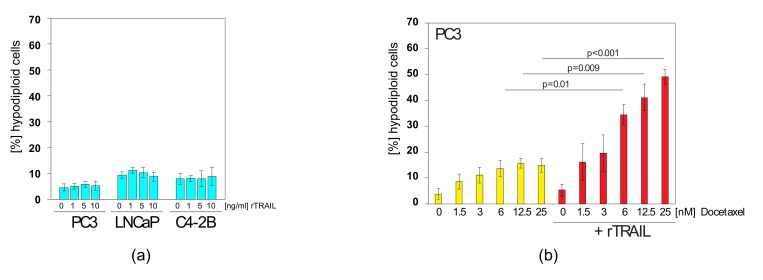
Docetaxel can sensitise TRAIL-resistant prostate cancer cells to the apoptosis-inducing effects of TRAIL. (**a**) PC3, LNCaP and C4-2B cells were treated with increasing concentrations of rTRAIL before apoptosis was measured. (**b**) PC3 cells were treated with increasing concentrations of docetaxel as indicated or treated with docetaxel/TRAIL mixes. TRAIL was used at 5 ng/mL. (**c**) LNCaP cells were treated with increasing concentrations of docetaxel as indicated or treated with docetaxel/TRAIL mixes. TRAIL was used at 5 ng/mL. (**d**) C4-2B cells were treated with increasing concentrations of docetaxel as indicated or treated with docetaxel/TRAIL mixes. TRAIL was used at 5 ng/mL. Values represent mean ± SE.

**Figure 5 cancers-11-00568-f005:**
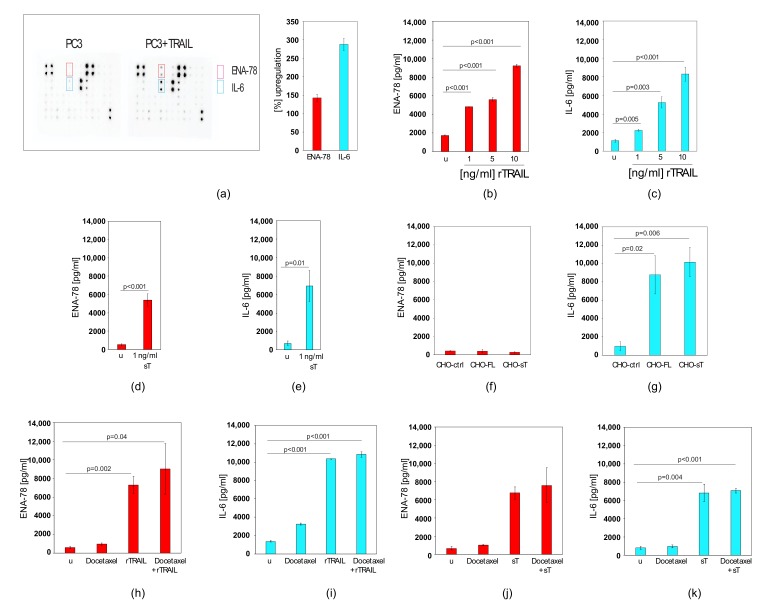
TRAIL induces the expression of CXCL5/ENA-78 and IL-6 that cannot be prevented by docetaxel co-treatment. (**a**) Cytokine arrays of untreated PC3 cells and PC3 cells treated with rTRAIL. The diagram on the right shows the quantification of the factors that were induced by rTRAIL (>50%). (**b**) CXCL5/ENA-78 ELISA results of supernatants from PC3 cells (u) and PC3 cells treated with increasing concentrations of rTRAIL. (**c**) IL-6 ELISA results of supernatants from PC3 cells (u) and PC3 cells treated with increasing concentrations of rTRAIL. (**d**) CXCL5/ENA-78 ELISA results of supernatants from PC3 cells (u) and PC3 cells treated with 1 ng/mL sTRAIL from MSCs. (**e**) IL-6 ELISA results of supernatants from PC3 cells (u) and PC3 cells treated with 1 ng/mL sTRAIL from MSCs. (**f**) CXCL5/ENA-78 ELISA results of supernatants from PC3 cells mixed with control transfected CHO cells (CHO-ctrl), PC3 cells mixed with FL-TRAIL expressing CHO cells (CHO-FL) and PC3 cells mixed with sTRAIL expressing CHO cells (CHO-sT). (**g**) IL-6 ELISA results of supernatants from PC3 cells mixed with control transfected CHO cells (CHO-ctrl), PC3 cells mixed with FL-TRAIL expressing CHO cells (CHO-FL) and PC3 cells mixed with sTRAIL expressing CHO cells (CHO-sT). (**h**) CXCL5/ENA-78 ELISA results of supernatants from untreated PC3 cells (u), PC3 cells treated with docetaxel, treated with rTRAIL or treated with a mix of docetaxel and rTRAIL. (**i**) IL-6 ELISA results of supernatants from untreated PC3 cells (u), PC3 cells treated with docetaxel, treated with rTRAIL or treated with a mix of docetaxel and rTRAIL. (**j**) CXCL5/ENA-78 ELISA results of supernatants from untreated PC3 cells (u), PC3 cells treated with docetaxel, treated with sTRAIL from MSCs or treated with a mix of docetaxel and sTRAIL. sTRAIL was used at 1 ng/mL, docetaxel was used at 12.5 nM. (**k**) IL-6 ELISA results of supernatants from untreated PC3 cells (u), PC3 cells treated with docetaxel, treated with sTRAIL or treated with a mix of docetaxel and sTRAIL. sTRAIL was used at 1 ng/mL, docetaxel was used at 12.5 nM. Values represent mean ± SE.

**Figure 6 cancers-11-00568-f006:**
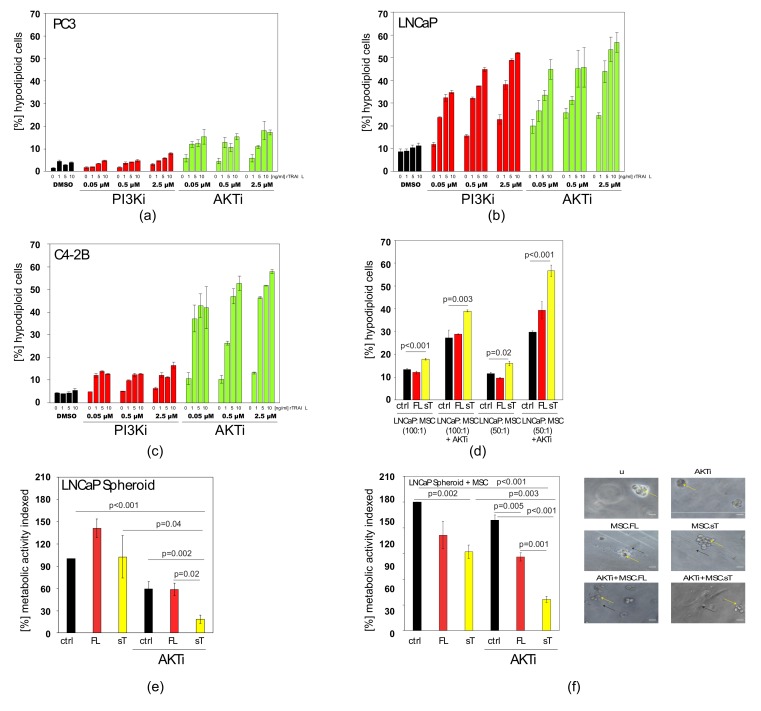
AKTi can sensitise prostate cancer cells to TRAIL and block cytokine production. (**a**) Apoptosis was measured in PC3 cells that were treated with increasing concentrations of PI3Ki or AKTi and co-treated with increasing concentrations of rTRAIL. DMSO is a vehicle control. (**b**) Apoptosis was measured in LNCaP cells that were treated with increasing concentrations of PI3Ki or AKTi and co-treated with increasing concentrations of rTRAIL. DMSO is a vehicle control. (**c**) Apoptosis was measured in C4-2B cells that were treated with increasing concentrations of PI3Ki or AKTi and co-treated with increasing concentrations of rTRAIL. DMSO is a vehicle control. (**d**) Apoptosis measurement of LNCaP cells mixed with control MSCs (ctrl), MSC.FL-TRAIL (FL) or MSC.sTRAIL (sT) at different ratios (50:1; 100:1) either with or without AKTi. (**e**) LNCaP cells were grown as 3D spheroids and then treated with supernatants from control MSCs (ctrl), full-length TRAIL (FL) MSCs and sTRAIL (sT) MSCs plus AKTi, before cell viability was assessed. (**f**) LNCaP cells were grown as spheroids in Matrigel and then treated with control MSCs (ctrl), MSC.FL-TRAIL (FL) and MSC.sTRAIL (sT) plus AKTi, before cell viability was assessed. The right panel shows representative images. Spheroids are indicated by yellow arrows, MSCs by black arrows. Scale bar is 30 μm. (**g**) Measurements of survival of untreated PC3 cells (u), PC3 cells treated with AKTi, treated with rTRAIL or treated with a mix of AKTi and rTRAIL. A representative image of these measurements is shown on the right side. (**h**) CXCL5/ENA-78 ELISA results of supernatants of untreated PC3 cells (u), PC3 cells treated with AKTi, treated with rTRAIL or treated with a mix of AKTi and rTRAIL. (**i**) IL-6 ELISA results of supernatants of untreated PC3 cells (u), PC3 cells treated with AKTi, treated with rTRAIL or treated with a mix of AKTi and rTRAIL. (**j**) CXCL5/ENA-78 ELISA results of supernatants of untreated PC3 cells (u), PC3 cells treated with AKTi, treated with sTRAIL (sT) from MSCs or treated with a mix of AKTi and sTRAIL (sT). (**k**) IL-6 ELISA results of supernatants of untreated PC3 cells (u), PC3 cells treated with AKTi, treated with sTRAIL (sT) or treated with a mix of AKTi and sTRAIL (sT). Values represent mean ± SE.

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
