# Peer review of "MSC.sTRAIL Has Better Efficacy than MSC.FL-TRAIL and in Combination with AKTi Blocks Pro-Metastatic Cytokine Production in Prostate Cancer Cells"

_cancers, 2019, doi:10.3390/cancers11040568_

Reviewer 1 Report

The study by Mohr et al. address interesting questions regarding the use of mesenchymal stem cells (MSC) transfected with TRAIL for tumor treatment. The study is well developed and the methodology used is adequate. However, the interpretation of the data obtained is rather confusin.

1) The title is completely misleading. The authors demostrate that MSC transfected with soluble TRAIL (sTRAIL) are more efficient against tumors than MSC transfected with full-lenght TRAIL (FL-TRAIL). They demonstrate that this is due to the fact that sTRAIL is secreted from MSC cells, but FL-TRAIL is not. Hence, the authors do not demonstrate that "Soluble TRAIL has better efficacy than full-length membrane-bound TRAIL...", as they state in the title and also in the abstract and in the rest of the paper, they just demonstrate that, since FL-TRAIL is not secreted from MSC cells, their supernantants are without activity, while MSC cellls transfected with sTRAIL do secrete the molecule, and it is active in the supernantant. In fact, it has been clearly demonstrated that membrane-bound TRAIL is much more efficient in signaling cell death than soluble TRAIL, since oligomerization of pre-assembled receptor trimers is needed (Clancy et al., PNAS 102: 18099, 2005; Chan et al., Cytokine 37: 101, 2007), and this has been also demonstrated in the context of anti-tumor activity, both in vrito and in vivo (recently revised in De Miguel et al., Cell Death Differ. 23: 733, 2016). FL-TRAIL could be only secreted from MSC associated with extracellular vesicles, and it seems that this does not happen in the experimental conditions of the present study. It has been described that MSC secrete exosomes, but only one study show that transfected TRAIL is secreted in exosomes from MSC (Shamili et al., Int. J. Pharm. 549: 218, 2018).

Hence, the title and abstract should be changed to avoid misleading interpretations of the data obtained, the considerations indicated should be taken into account and discussed in the paper and the references indicated cited.

2) Figures are too small, and especially their lettering. The size of lettering and the size of figures should be increased.

Author Response

Reviewer 1 (all changes in the manuscript are highlighted in yellow for reviewer 1)

1.      The title is completely misleading. The authors demostrate that MSC transfected with soluble TRAIL (sTRAIL) are more efficient against tumors than MSC transfected with full-lenght TRAIL (FL-TRAIL). They demonstrate that this is due to the fact that sTRAIL is secreted from MSC cells, but FL-TRAIL is not. Hence, the authors do not demonstrate that "Soluble TRAIL has better efficacy than full-length membrane-bound TRAIL...", as they state in the title and also in the abstract and in the rest of the paper, they just demonstrate that, since FL-TRAIL is not secreted from MSC cells, their supernantants are without activity, while MSC cellls transfected with sTRAIL do secrete the molecule, and it is active in the supernantant. In fact, it has been clearly demonstrated that membrane-bound TRAIL is much more efficient in signaling cell death than soluble TRAIL, since oligomerization of pre-assembled receptor trimers is needed (Clancy et al., PNAS 102: 18099, 2005; Chan et al., Cytokine 37: 101, 2007), and this has been also demonstrated in the context of anti-tumor activity, both in vrito and in vivo (recently revised in De Miguel et al., Cell Death Differ. 23: 733, 2016). FL-TRAIL could be only secreted from MSC associated with extracellular vesicles, and it seems that this does not happen in the experimental conditions of the present study. It has been described that MSC secrete exosomes, but only one study show that transfected TRAIL is secreted in exosomes from MSC (Shamili et al., Int. J. Pharm. 549: 218, 2018).

Hence, the title and abstract should be changed to avoid misleading interpretations of the data obtained, the considerations indicated should be taken into account and discussed in the paper and the references indicated cited.

Response: We did not try to imply that mechanistically, sTRAIL as such has better apoptosis-inducing effects than membrane bound FL-TRAIL. The objective of these experiments was to compare FL-TRAIL and sTRAIL in settings that mimic MSC-based cell therapy, i.e. comparably few infiltrated MSCs against a majority of cancer cells. However, we do recognise that our title and wording can be misleading. Therefore, we modified the title and the abstract, have added the suggested references and discussed the points in our manuscript.

As mentioned above, our results do not question that FL-TRAIL has apoptosis-inducing effects, and in a ‘one-on-one’ situation can exert more apoptosis than sTRAIL. This is, because, as the reviewer pointed out, ligand-independent formation of trimeric receptor complexes plays a crucial role in the function and signalling of death receptors, therefore preferring trimerised ligand, which can be better forced in the membrane. In order to translate these advantages from membrane-bound TRAIL to the sTRAIL system, the latter is linked to an ILZ domain forcing trimerization, thereby increasing its activity.

Changes: We made changes to the title, abstract and throughout the manuscript that are highlighted in yellow in the revised manuscript version named ‘markup’. Furthermore, we added a discussion on pre-assembled receptor trimers using the indicated references that we cited.

2.       Figures are too small, and especially their lettering. The size of lettering and the size of figures should be increased.

Response: We agree with the reviewer and have enlarged all figures and labels, where appropriate.

Changes: Figures were made larger and the size of labels inside of figures was also increased.

Reviewer 2 Report

The manuscript submitted by Mohr et al. deals with a detail analysis of TRAIL-induced signaling originated from mouse or human mesenchymal stem cells (MSC) transduced either with the cytoplasmic membrane associated FL human TRAIL or with modified TRAIL containing FIB signal peptide, furin cleavage site, isoleucine zipper (ILZ) trimerization domain and shortened extracellular domain of human TRAIL (AAs 114-281) named soluble TRAIL (sT). The authors found out that in contrast to the membrane bound FL TRAIL is the soluble TRAIL more effective in inducing apoptosis of tested cancer cells and that it also better cooperates with other anti-cancer drugs either approved as docetaxel or tested as PI3K and Akt inhibitors. Thus considering transduced MSC as possible anti-cancer treatment, the authors proposed that using MSCs transduced with ., soluble recombinant TRAIL would be significantly more effective approach than just overexpressed membrane-bound TRAIL. Moreover, the authors document that the concurrent TRAIL-induced expression of tumor growth-enhancing factors CXCL5 and IL-6 is suppressed in TRAIL+ Akt inhibitor MK-2206-treated prostate cancer cells. In general taking in account availability (sT accumulates), accessibility (better access via diffusion) and likely efficacy (enhanced forced trimerized TRAIL)  of membrane-bound wt TRAIL vs soluble modified TRAIL is not too surprising that the membrane-bound TRAIL was inferior to the soluble one in inducing apoptosis of cancer cells.

Questions and comments:

1.    As the membrane-bound and the soluble TRAIL could in addition to their accesibility also differ in efficacy it would be instrumental to test more relevant comparison such as membrane-bound ILZ-TRAIL vs. wt TRAIL and under more appropriate conditions – i.e. seeding fluorescent-labeled cancer cells onto a monolayer of TRAIL-expressing MSCs (membrane-bound or secreted) instead of using transwell.

2.    In the analysis of TRAIL-induced expression of chemokines/cytokines was used just either recombinant (RaD – just plain shortened extracellular part of hTRAIL AAs 114-281) or the cell culture supernatant containing soluble ILZ-TRAIL. Thus it would be worth testing whether the membrane bound TRAIL will provide the same cytokine-expression profile as both soluble versions did.

3.    There is likely a typo in Fig.6 as h+i and j+k graphs refer to PC3 cells.

Author Response

Reviewer 2 (all changes in the manuscript are highlighted in green for reviewer 2)

1.      As the membrane-bound and the soluble TRAIL could in addition to their accesibility also differ in efficacy it would be instrumental to test more relevant comparison such as membrane-bound ILZ-TRAIL vs. wt TRAIL and under more appropriate conditions – i.e. seeding fluorescent-labeled cancer cells onto a monolayer of TRAIL-expressing MSCs (membrane-bound or secreted) instead of using transwell.

Response: The objective of our study was to compare the currently most frequently discussed and used versions of TRAIL in the context of cell therapy, which are sTRAIL linked to an ILZ domain and full-length, membrane-bound TRAIL. However, it is clear that there are other versions of TRAIL that are being tested as reviewed in De Miguel et al., Cell Death Differ. 23: 733, 2016, and as pointed out by Reviewer 1. This paper is now cited in our manuscript.

In addition to supernatant transfer and transwell experiments we had mixed TRAIL expressing cells (CHO cells and MSCs) and presented the results for apoptosis and cytokine expression in Figures 3i, j, k; 5f, g and 6d, f. The objective of these experiments was to compare FL-TRAIL and sTRAIL in settings that mimic MSC-based cell therapy, i.e. comparably few infiltrated MSCs against a majority of cancer cells.

Changes: Cited paper and inserted new Figure 5f and g. 

2.      In the analysis of TRAIL-induced expression of chemokines/cytokines was used just either recombinant (RaD – just plain shortened extracellular part of hTRAIL AAs 114-281) or the cell culture supernatant containing soluble ILZ-TRAIL. Thus it would be worth testing whether the membrane bound TRAIL will provide the same cytokine-expression profile as both soluble versions did.

Response: In response to the reviewer’s comment we analysed Chinese Hamster Ovary (CHO) cells expressing FL-TRAIL or sTRAIL that we mixed with PC3 cells. We chose CHO cells for this experiment to avoid any interference from MSC-derived factors. The results show that for both FL-TRAIL and sTRAIL IL-6 levels increased, but in contrast to recombinant TRAIL or cell-free sTRAIL the CXCL5/ENA-78 levels in the same experiment remained unchanged. We do not have a mechanistic explanation for this effect, but the results point to additional benefits of cell delivered TRAIL that are worthy to be studied in more detail in the future.

Changes: We added the ELISA results in Figure 5 (new Figure 5f and g) and added corresponding descriptions in the Methods, Results and Discussion part. These changes are highlighted in green in the revised manuscript version named ‘markup’.

The other figures in Figure 5 were changed as follows:

Old 5f became 5h

Old 5g became 5i

Old 5h became 5j

Old 5i became 5k 

3.      There is likely a typo in Fig.6 as h+i and j+k graphs refer to PC3 cells.

Response:  We checked the graphs and the corresponding text. They do contain data and descriptions of results from PC3 cells.  In this experiment, we investigated the effect of AKTi upon TRAIL-induced cytokine expression in PC3 cells.  As far as we can see, the labelling is correct for this figure.

Changes: No changes to manuscript.

Reviewer 3 Report

MS# Cancers-467654

The authors prepared MSC lines that express full-length membrane-bound TRAIL or soluble-form TRAIL and then compared their anti-tumor effects on a human colon cancer cell line HT-29 and three human prostate cancer cell lines. Based on their results, they conclude that soluble TRAIL has better efficacy than full-length membrane-bound TRAIL, and that an AKT inhibitor but not Docetaxel can suppress the production of pro-metastatic cytokines including IL-6 and CXCL5 by TRAIL-treated prostate cancer cells. The reviewer supposes that this study contains interesting information for the readers but several point should be clarified.

Specific comments:

1)  Some figures such as Fig 3k (right), Fig 5a (left), and Fig 6f (right) are too small.

2)  They conclude that soluble TRAIL has better efficacy than full-length membrane-bound TRAIL However, in Fig 3i, j, and k, the ratios of cancer cells/MSC are 5, 10, and 100. For the reviewer, it seems simply because full-length membrane-bound TRAIL could not exert anti-tumor effects effectively simply because membrane-bound TRAIL needs cell contact with cancer cells but the numbers of MSC were considerably small compared with cancer cells. In contrast, soluble TRAIL can freely bind to death receptors on cancer cells. More numbers of MSC could decrease the difference in therapeutic efficacy between full-length TRAIL/MSC and soluble-form TRAIL/MSC.

3)  All experiments were done only in vitro. Additional experiments with xenografted in vivo model are desirable.

Author Response

Reviewer 3

1.      Some figures such as Fig 3k (right), Fig 5a (left), and Fig 6f (right) are too small.

Response: We agree with the reviewer and have enlarged all figures and labels, where possible, focusing on the figures mentioned above.

Changes: Figures were made larger and the size of labels inside of figures were also increased. 

2.      They conclude that soluble TRAIL has better efficacy than full-length membrane-bound TRAIL However, in Fig 3i, j, and k, the ratios of cancer cells/MSC are 5, 10, and 100. For the reviewer, it seems simply because full-length membrane-bound TRAIL could not exert anti-tumor effects effectively simply because membrane-bound TRAIL needs cell contact with cancer cells but the numbers of MSC were considerably small compared with cancer cells. In contrast, soluble TRAIL can freely bind to death receptors on cancer cells. More numbers of MSC could decrease the difference in therapeutic efficacy between full-length TRAIL/MSC and soluble-form TRAIL/MSC.

Response: The objective of these experiments was to compare FL-TRAIL and sTRAIL in settings that mimic cell therapeutic settings, i.e. comparably few infiltrated MSCs against a majority of cancer cells. Therefore, we used cancer cells/MSC ratios of 5:1 to 100:1. Our results do not question that FL-TRAIL has apoptosis-inducing effects, and in a ‘one-on-one’ situation can exert at least as much apoptosis as sTRAIL. Reviewer 1 raised a similar point, so more details can be found in our response there.

Changes: No changes to manuscript.

3.      All experiments were done only in vitro. Additional experiments with xenografted in vivo model are desirable.

Response: We previously showed that systemically administered sTRAIL-loaded MSCs have anti-tumour effects in colorectal and pancreatic xenograft models. Thus, the approach works in principle. Here, we wanted to compare FL-TRAIL to sTRAIL in different models and analyse the effects on cytokine production. In addition, we conducted experiments with 3D- cultures of prostate cancer cells. Therefore, we think that xenograft studies would go beyond the scope of the present manuscript.

Changes: No changes to manuscript.

Reviewer 4 Report

The manuscript demonstrated the effects of soluble TRAIL in prostate cancer cells and that AKTi affect cytokine levels induced by TRAIL. Several points as outlined below need to be clarified and/or addressed to further boost the significance of the findings.

Major

1. It looks like that colo205 cells are more sensitive to TRAIL compared to HT-29 cells. However, authors should explain why apoptosis was induced in colo205 cells in conjunction with the secretion status of TRAIL from MSC.FL-TRAIL.

2. Authors demonstrated that docetaxel+TRAIL treatment did not result in the downregulation of ENA-78 and IL-6 even though docetaxel treatment increases TRAIL effects. In addition, they showed the Akti effects on these cytokine levels. Some questions can be derived from those results. 1) TRAIL treatment affect AKT activity? , 2) Does docetaxel not affect akt activity?, 3) How authors can explain docetaxel effects on TRAIL and 4) what can be expected when docetaxel is treated with sTRAIL and AKTi?

3. Instead of using pharmacological inhibitors, authors need to use RNAi approaches.

Minor

1. Please take this opportunity to check any typos and grammar throughout the manuscript.

2. Line 61: Sentence is not clear to explain the relationship between MOMP and XIAP.

3. Although authors demonstrated the secretion status of FL-TRAIL in Fig. 3a and b, it is necessary to add FL-TRAIL data in Fig. 3c to more precisely compare the secreted levels.

4. What % of hypodiploid cells were detected for AT, BM, and UC?

5. Representative cell images in Fig. 3 are too small. In addition, any representative cell images of HT-29 from co-culture? And add scale bars.

6. What concentration of rTRAIL was used in Fig. 4b, c, d?

7. For Fig. 5h and i, the amount of soluble TRAIL is 1 ng/ml? also, indicate the concentration of other treatment reagents.

8. Adjust the scale of array data for the reader and also mention the unaffected cytokines by TRAIL treatment in the result section and/or in the Fig 5.

9. Add bar graphs showing % of apoptotic cells of TRAIL+vehicle treatment in Fig. 6a, b, c.

10. Re-arrange the arrows and scale up the images of spheroids in Fig. 6.

Author Response

Reviewer 4 (all changes in the manuscript are highlighted in turquoise for reviewer 4)

Major

1.      It looks like that colo205 cells are more sensitive to TRAIL compared to HT-29 cells. However, authors should explain why apoptosis was induced in colo205 cells in conjunction with the secretion status of TRAIL from MSC.FL-TRAIL.

Response: Yes, Colo205 are more sensitive to TRAIL than HT29 cells. Thus, while in Colo205 cells significant apoptosis could still be measured at a cancer cells/MSC.FL-TRAIL ratio of 10:1, this was not detectable when HT29 were mixed with MSC.FL-TRAIL at the same ratio. These results show that MSC.FL-TRAIL has cancer cell killing effects when applied in relatively high numbers on very sensitive cells like Colo205 cells, but the effect sharply declines when cancer cells are not as responsive (e.g. HT29 cells). This is in contrast to MSC.sTRAIL, where we see effects at higher ratios, i.e. up to 100:1 for Colo205 cells. Our goal was to compare FL-TRAIL and sTRAIL in settings that resemble the situation when MSCs are systemically administered, i.e. comparably few infiltrated MSCs against a majority of cancer cells. Therefore, we used cancer cells/MSC ratios of 5:1 to 100:1 in mixing experiments with Colo205 cells. As mentioned above, at all ratios sTRAIL had better effects than FL-TRAIL, and at the higher ratio (100:1) MSC.FL-TRAIL failed to induce apoptosis, whereas MSC.sTRAIL still exerted cell death rates of around 35% in Colo205 cells.

Changes: No changes to the manuscript

2.      Authors demonstrated that docetaxel+TRAIL treatment did not result in the downregulation of ENA-78 and IL-6 even though docetaxel treatment increases TRAIL effects. In addition, they showed the Akti effects on these cytokine levels. Some questions can be derived from those results. 1) TRAIL treatment affect AKT activity? , 2) Does docetaxel not affect akt activity?, 3) How authors can explain docetaxel effects on TRAIL and 4) what can be expected when docetaxel is treated with sTRAIL and AKTi?

Response: We carried out additional experiments to measure 1) AKT activity after TRAIL treatment, 2) AKT activity after Docetaxel treatment as well as Docetaxel/TRAIL co-treatments. The results show that TRAIL and Docetaxel treatments did not affect AKT phosphorylation/activation, whereas treatment with AKTi showed a substantial reduction (Supplementary Figure 1b). 3) Furthermore, we measured TRAIL-receptor expression after Docetaxel treatment and did not observe a rise in the levels of TRAIL-R1 or TRAIL-R2 on the surface of treated prostate cancer cells (Supplementary Figure 1a). Thus, a different mechanism must account for the TRAIL sensitisation afforded by Docetaxel. Yoo et al found that in C4-2B cells this was mediated mainly by phosphorylation of Bcl-2 by JNK activation. This paper was cited (reference 88), but is now discussed in more detail. 4) When we co-treated with Docetaxel sTRAIL and AKTi we found only a small further decrease in survival, but no further improvement in relation to cytokine production (Supplementary Figure 1c, d and e).

Changes: We added the results of these experiments to a new Figure (Supplementary Figure 1) Furthermore, we added corresponding descriptions in the Methods and Results part. These changes are highlighted in turquoise in the revised manuscript version named ‘markup’.

3.      Instead of using pharmacological inhibitors, authors need to use RNAi approaches.

Response: As we consider using these approaches in vivo in the future, RNAi approaches are not helpful in this direction. We rather concentrated on the effects of existing inhibitors that are being tested clinically. The elucidation of detailed mechanistic mechanisms was not the underlying goal for this project and go beyond the scope of the present manuscript.

Minor

1.      Please take this opportunity to check any typos and grammar throughout the manuscript.

Response: We re-checked the manuscript for typos and grammar and made a number of corrections.

Changes: Several typos and smaller errors have been corrected. 

2.      Line 61: Sentence is not clear to explain the relationship between MOMP and XIAP.

Response: We broke down the sentence into two and reworded them to increase clarity of the sentence.

Changes: Line 61-66: “There, the pro-apoptotic proteins are believed to form pores giving rise to mitochondrial outer membrane permeabilisation (MOMP). MOMP results in the release of cytochrome c, Smac/DIABLO and other pro-apoptotic factors into the cytosol. Cytochrome c amplifies the apoptotic signal emitted by activated caspase-8 via additional caspase-9 activation in the apoptosome, while Smac/DIABLO releases the XIAP block on caspase-9 and executioner caspases, thereby also strengthening apoptosis [22-24]”. 

3.      Although authors demonstrated the secretion status of FL-TRAIL in Fig. 3a and b, it is necessary to add FL-TRAIL data in Fig. 3c to more precisely compare the secreted levels.

Response: We carried out additional experiments comparing sTRAIL and FL-TRAIL in the different MSC types and added the new ELISA results to Fig 3c.

Changes: A revised Figure 3c replaced the old Figure 3c. 

4.      What % of hypodiploid cells were detected for AT, BM, and UC?

Response: We added the apoptosis levels for AT, BM and UC MSCs as measured by hypodiploidy assays to Figure 3d.

Changes: A revised Figure 3d replaced the old Figure 3d. 

5.      Representative cell images in Fig. 3 are too small. In addition, any representative cell images of HT-29 from co-culture? And add scale bars.

Response: We have increased the sizes of the images in Figure 3 and added scale bars.

Changes: A revised Figure 3k replaced the old Figure 3k. 

6.      What concentration of rTRAIL was used in Fig. 4b, c, d?

Response: We added this information to the corresponding figure legend (Figure 4).

Changes: The rTRAIL concentrations are now provided in the corresponding figure legend of Figure 4. 

7.      For Fig. 5h and i, the amount of soluble TRAIL is 1 ng/ml? also, indicate the concentration of other treatment reagents.

Response: We added this information to the corresponding figure legend.

Changes: The concentrations of the other treatment reagent are now provided in the figure legend for Figure. Other detailed concentration information are provided in the Methods part. 

8.      Adjust the scale of array data for the reader and also mention the unaffected cytokines by TRAIL treatment in the result section and/or in the Fig 5.

Response: We adjusted the scale for the array data and also listed the unaffected cytokines in Supplementary Figure 2.

Change: We added a supplementary figure showing the cytokine array configuration and show the increased (highlighted) as well as unchanged/decreased factors (Supplementary Figure 2). 

9.      Add bar graphs showing % of apoptotic cells of TRAIL+vehicle treatment in Fig. 6a, b, c.

Response: We added the results for vehicle treatment in the diagram in Figure 6a, b and c.

Changes: A revised Figure 6a, b, c replaced the old Figure 6a, b, c. 

10.  Re-arrange the arrows and scale up the images of spheroids in Fig. 6.

Response: We re-arranged the arrows and increased the size of Figure 6d.

Changes: A revised Figure 6d replaced the old Figure 6d

Round  2

Reviewer 1 Report

The authors made changes according to the suggestions made in the first revision.

Reviewer 2 Report

I do not have any further comments to this manuscript and thus I do recommend accepting it for publication in the journal.

Reviewer 4 Report

The authors have significantly improved the manuscript in an impressively short time. All required experiments and descriptions have been included. Confusion about some structure descriptions has been resolved with an appropriate way. I do not have any further concerns.